# Dietary Supplementation with 20-Hydroxyecdysone Ameliorates Hepatic Steatosis and Reduces White Adipose Tissue Mass in Ovariectomized Rats Fed a High-Fat, High-Fructose Diet

**DOI:** 10.3390/biomedicines11072071

**Published:** 2023-07-23

**Authors:** Jariya Buniam, Piyachat Chansela, Jittima Weerachayaphorn, Vitoon Saengsirisuwan

**Affiliations:** 1Princess Srisavangavadhana College of Medicine, Chulabhorn Royal Academy, Bangkok 10210, Thailand; jariya.bun@cra.ac.th; 2Department of Anatomy, Phramongkutklao College of Medicine, Bangkok 10400, Thailand; 3Department of Physiology, Faculty of Science, Mahidol University, Bangkok 10400, Thailand

**Keywords:** 20-hydroxyecdysone, metabolic dysfunction-associated fatty liver disease (MAFLD), ovariectomy, high-fat high-fructose diet, obesity

## Abstract

Metabolic dysfunction-associated fatty liver disease (MAFLD) is defined as hepatic steatosis in combination with overweight, diabetes, or other metabolic risk factors. MAFLD affects a significant number of the global population and imposes substantial clinical and economic burdens. With no approved pharmacotherapy, current treatment options are limited to diet and exercise. Therefore, the development of medicines for MAFLD treatment or prevention is necessary. 20-Hydroxyecdysone (20E) is a natural steroid found in edible plants and has been shown to improve metabolism and dyslipidemia. Therefore, it may be useful for MAFLD treatment. Here, we aimed to determine how dietary supplementation with 20E affects fat accumulation and lipogenesis in the liver and adipose tissue of ovariectomized rats fed a high-fat, high-fructose diet (OHFFD). We found that 20E reduced hepatic triglyceride content and visceral fat deposition. 20E increased the phosphorylation of AMP-activated protein kinase and acetyl CoA carboxylase while reducing the expression of fatty acid synthase in the liver and adipose tissue. Additionally, 20E increased hepatic expression of carnitine palmitoyltransferase-1 and reduced adipose expression of sterol regulatory element-binding protein-1. In conclusion, 20E demonstrated beneficial effects in rats with OHFFD-induced MAFLD. These findings suggest that 20E may represent a promising option for MAFLD prevention or treatment.

## 1. Introduction

“Metabolic dysfunction-associated fatty liver disease” (MAFLD), a potential replacement for the term “non-alcoholic fatty liver disease”, is defined on the basis of evidence of hepatic steatosis, in addition to one of the following three criteria: the presence of overweight/obesity, the presence of type 2 diabetes mellitus, and evidence of metabolic dysregulation [1,2]. MAFLD now affects more than 1 billion people worldwide and is the cause of substantial healthcare and economic burdens worldwide [1,2,3]. The surge in the prevalence of MAFLD is, to a large extent, the result of the consumption of unhealthy diets, such as high-calorie diets, which are rich in processed foods and saturated fat, and include beverages sweetened with fructose [4]. Furthermore, accumulating evidence indicates that the prevalence of metabolic dysfunction is higher when the concentrations of ovarian hormones dwindle, such as in menopausal women [5,6,7,8]. Previous studies have demonstrated that ovariectomized rats fed a high-fat, high-fructose diet (OHFFD) develop multiple abnormalities, including hepatic steatosis [9,10], higher body mass, impaired glucose tolerance, and insulin resistance [9,10,11]. Thus, this animal model of OHFFD mimics MAFLD in menopausal women who are consuming high-calorie diets. Although lifestyle intervention, consisting of calorie restriction and exercise, remains the foundation of the management of MAFLD, it is often difficult to sustain such lifestyle changes. Therefore, an effective pharmacological treatment for MAFLD is sought.

Pioglitazone (PIO), a peroxisome proliferator-activated receptor-gamma (PPAR-γ) agonist, is used to ameliorate some aspects of MAFLD [12,13]. Nevertheless, the prolonged use of PIO may have undesirable effects, including edema, pulmonary edema, hepatic steatosis, and congestive heart failure [13]. Therefore, the development of substances for the treatment of MAFLD should prioritize candidates with few side effects [14]. Accordingly, there has been a great deal of research into the use of nutraceuticals or plant-derived supplements for the treatment of hepatic steatosis and to reduce white adipose tissue accumulation. Interestingly, 20-hydroxyecdysone (20E), a polyhydroxylated insect steroid hormone, is also found in certain plant species, such as spinach and quinoa, which have been referred to as “superfoods” because they contain higher contents of bioactive components than equivalent foods [15,16,17,18].

Although plants containing 20E are used in traditional medicines in Asia and Western countries, their bioactivities have not been thoroughly studied. During recent years, increasing attention has been paid to 20E because of its effects on metabolism. Specifically, it has been shown to improve glucose metabolism and insulin sensitivity in obese rats [19,20,21] and ameliorate dyslipidemia [11,18], but its effects on visceral and hepatic fat accumulation have not been investigated. Therefore, in the present study, we aimed to investigate the effect of long-term supplementation with 20E on fat accumulation in the liver and white adipose tissue and determine how 20E modulates lipogenesis in OHFFD rats, which mimic MAFLD in humans. A better understanding of the mechanisms responsible for the beneficial effects of 20E in these two important metabolic organs would underpin the use of a nutraceutical such as 20E for the treatment of symptoms of MAFLD and the prevention of its progression.

## 2. Materials and Methods

### 2.1. Animal Studies

Eight-week-old female Sprague-Dawley rats, weighing approximately 180–200 g, were obtained from the National Laboratory Animal Center, Thailand. The rats were housed in the animal facility under controlled temperature conditions, at 22 °C, and under a 12:12 h light-dark cycle. After an acclimatization period, we designated the experimental groups as follows: (i) sham-operated rats fed a control diet (Sham); (ii) ovariectomized (OVX) rats fed a high-fat, high-fructose diet (OHFFD); (iii) OVX rats fed a high-fat, high-fructose diet and administered 20E (OHFFD + 20E) at a dose of 5, 10, or 20 mg/kg body mass; and OVX rats fed a high-fat, high-fructose diet and administered PIO at a dose of 10 mg/kg body mass (OHFFD + PIO). The number of rats used (n = 6 per group) was calculated using Minitab 21.2.0 Software (Minitab Inc., State College, PA, USA). The body masses of the rats were recorded at the start of the study. At 10 weeks of age, ovariectomy or sham surgery was performed as previously described [22]. Briefly, before surgery, rats were under deep anesthesia by intraperitoneally injecting with a combination of Tiletamine-zolazepam (Zoletil 50, Virbac Laboratories, Carros, France) at a dosage of 25 mg/kg and xylazine (Thai Meiji Pharmaceutical, Bangkok, Thailand) at a dosage of 3–5 mg/kg. Ovariectomy was performed on anesthetized animals through bilateral 1.5-cm paralumbar incisions. The distal part of the fallopian tubes was ligated before both ovaries were removed. Subsequently, the skin was sutured and cleaned with 70% ethanol and povidone-iodine, and vital signs were carefully observed until rats were recovered from anesthesia. The OVX rats were allowed to recover for 7 days after surgery. After 7 days of recovery from ovariectomy, sham-operated rats were fed a control diet (TD08806, 10% kcal as fat; Envigo (formerly Harlan Teklad), Indianapolis, IN, USA) and provided with reverse osmosis-derived water. The OHFFD rats were fed a high-fat diet (TD06414, 60% kcal as fat; Envigo) and provided with 30% w/v fructose in their drinking water for 12 weeks ad libitum to create the MAFLD model. After 4 weeks of the diet regimen, the OHHFD rats were administered 20E isolated and purified from the tropical plant Vitex glabrata as described previously [23] at a dose of 5, 10, and 20 mg/kg body mass or PIO (10 mg/kg) diluted in 25% propylene glycol by oral administration daily between 09:00 and 10:00. The doses of 20E used in the study were based on previous studies [18,19,20,24]. Throughout the 12 weeks of the study, the food and water intake of the rats was measured three times a week. The total caloric intake of each animal was calculated using their mean daily food and water (or fructose solution) intake and the caloric contents of the respective diets. The animal protocol was approved by the Institutional Animal Care and Use committee of the Faculty of Science, Mahidol University (IACUC approval number: MUSC58-007-322) in accordance with the International Guiding Principles for Biomedical Research Involving Animals of the Council for International Organizations of Medical Sciences. The present study is reported in accordance with ARRIVE guidelines (https://arriveguidelines.org (accessed on 8 February 2021)).

### 2.2. Tissue Collection

At the end of the experiment, 15 h prior to euthanasia, the rats were food-restricted (4 g per rat), and the fructose solution was replaced with reverse osmosis-derived water. The rats were weighed and then anesthetized using an intraperitoneal injection of pentobarbital sodium (Nembutal; 75 mg/kg). The liver and visceral adipose tissue depots (mesenteric fat, retroperitoneal fat, and perigonadal fat) of the rats were excised and immediately weighed, and euthanasia was followed by rapid excision of the heart. Part of the liver was fixed in 10% neutral-buffered formalin, and the remainder of the liver and the adipose tissue depots were frozen in liquid nitrogen and stored at −80 °C until analyzed.

### 2.3. Histopathological Analysis

Formalin-fixed livers were embedded in paraffin, sectioned at a thickness of 5 µm (Leica Biosystems, Singapore), and processed for hematoxylin and eosin (H&E) staining. The histology of the liver was evaluated, and images were generated using an Olympus VS120 slide scanner. The images were analyzed using a Digital Pathology 3DHISTECH slide scanner (Olympus, Tokyo, Japan).

### 2.4. Liver Triglyceride Content

Liver triglyceride content was analyzed using a Triglyceride Quantification Kit (Abcam, Cambridge, UK), according to the manufacturer’s instructions.

### 2.5. Immunoblotting Analysis

Frozen rat liver tissue (50 mg) and periovarian white adipose tissue (50 mg) samples were homogenized using a Tissue Lyser LT (Qiagen, Hilden, Germany) in 500 μL aliquots of ice-cold RIPA lysis buffer (Thermo Scientific Inc., Waltham, MA, USA) containing a Halt protease and phosphatase inhibitor cocktail (Thermo Scientific). Liver and adipose tissue homogenates were centrifuged at 16,000× *g* and 4 °C for 20 min, and the protein concentrations of the supernatants were determined using a BCA protein assay kit (Pierce, Rockford, IL, USA). Lysates containing 50 µg of protein were separated using 8–10% SDS-PAGE and blotted onto 0.45 μm nitrocellulose membranes (Bio-Rad, Richmond, CA, USA). The blots were blocked using 5% Omniblok non-fat dried milk (AmericanBio, Kinderhook, NY, USA) or 5% bovine serum albumin (BSA) (Thermo Fisher Scientific) in tris-buffered saline (TBS) containing 0.1% Tween-20 for 2 h. The membranes were cut prior to incubating at 4 °C overnight with one of the following primary antibodies: AMPKα (1:800, Cell Signaling, Danvers, MA, USA), phospho-AMPKα (Thr^172^) (1:800, Cell Signaling), ACC (1:500, Cell Signaling), phospho-ACC (Ser^79^) (1:500, Cell Signaling), CPT-1 (1:500, Santa Cruz, Dallas, TX, USA), FAS (1:800, Cell Signaling), SREBP-1c (1:800; Santa Cruz), or GAPDH (1:3000, Cell Signaling). The blots were subsequently incubated with horseradish peroxidase (HRP)-conjugated goat anti-rabbit IgG (1:3000) or HRP-conjugated goat-anti mouse IgG secondary antibody (1:3000) at room temperature for 1 h. Protein bands were visualized using enhanced chemiluminescence (PerkinElmer Life Sciences) on a LI-COR C-DiGit Blot Scanner (LI-COR Biotechnology, Lincoln, NE, USA) and the band densities were quantified using Image Studio software. GAPDH was used as the reference protein, and the expression of each target protein was normalized to that of GAPDH.

### 2.6. Statistical Analysis

GraphPad Prism 8.0 (GraphPad Software, La Jolla, CA, USA) was used for data analysis. All data are expressed as mean ± standard error of the mean (SEM). The groups were compared using one-way ANOVA, followed by Tukey’s post-hoc test, as appropriate. The changes in body mass during the experimental period were analyzed using two-way ANOVA, followed by Tukey’s post-hoc test. Differences with *p* < 0.05 were considered to be statistically significant.

## 3. Results

### 3.1. 20-Hydroxyecdysone Reduces Body Mass Gain and Visceral Adipose Mass

The body mass changes during the 12 weeks of experimental diet consumption are shown in Figure 1A. We found significant differences in the mean body masses of the Sham and OHFFD rats, starting from the 2-week time point, and the OHFFD rats continued to gain weight until the end of the experiment. Of note, supplementation with 20E, particularly at the high dose (20 mg/kg), significantly reduced the body mass gain of the OHFFD rats from the 2-week time point and promoted the weight loss induced by OHFFD between weeks 6 and 12 of the study. However, there were no significant differences in the body masses of the OHFFD and OHFFD + PIO groups (Figure 1A). Although there was a difference in the total caloric intake of the Sham and OHFFD groups, there were no differences in the total caloric intakes of the OHFFD groups (Figure 1B). Moreover, the masses of the mesenteric, retroperitoneal, and periovarian fat depots were significantly higher in the OHFFD group than in the Sham group, and 20E supplementation significantly reduced all these masses to a similar extent to PIO (Figure 1C–E). Taken together, these results suggest that 20E reduces the excessive accumulation of white adipose tissue in OHFFD rats without affecting caloric intake.

### 3.2. 20-Hydroxyecdysone Reduces Hepatic Lipid Accumulation

The link between hepatic lipid accumulation and local insulin resistance is well established [25,26,27]. Chronic high fat, high fructose diet-feeding induces hepatic steatosis and reduces whole-body insulin sensitivity in ovariectomized rats [9]. Therefore, we analyzed the histology of the liver and measured the hepatic lipid contents of the rats to assess the effects of 20E supplementation on hepatic steatosis.

The histologic analysis showed no fat deposition in the liver of the Sham group, whereas the OHFFD rats showed marked macro- and micro-vesicular steatosis was reduced by 20E supplementation to a similar extent to PIO treatment (Figure 2A). Likewise, the OHFFD rats showed significantly higher hepatic triglyceride content (Figure 2B), and 20E supplementation significantly reduced this, as was the case for PIO treatment. These results suggest that 20E has an inhibitory effect on hepatic triglyceride accumulation, and therefore that it may protect against OHFFD-induced hepatic steatosis.

### 3.3. 20-Hydroxyecdysone Ameliorates Hepatic Steatosis by Activating AMPK Phosphorylation and Reducing the β-Oxidation of Fatty Acids

Hepatic fat accumulation occurs because of an imbalance between lipid availability and disposal created by an increase in lipid synthesis and a decrease in lipid oxidation [26,27]. In fatty liver disease, the hepatic uptake of fatty acids and de novo lipogenesis are upregulated, and the compensatory increase in fatty acid oxidation is insufficient to normalize lipid concentrations [25,27]. To determine how 20E modulates lipid metabolism and attenuates hepatic lipid accumulation, we measured the expression of the rate-limiting enzyme of fatty acid metabolism and its regulators (ACC and AMPKα), the rate-limiting enzyme of the fatty acid synthesis pathway (FAS), and the rate-limiting enzyme of fatty acid β-oxidation, carnitine palmitoyltransferase 1 (CPT-1) (Figure 3A–D and Figure 4A–E). As expected, the OHFFD group showed significantly higher expression of FAS and lower phosphorylation of AMPKαThr^172^ and ACC Ser^79^. By contrast, both 20E and PIO reduced the hepatic protein expression of fatty acid synthase (FAS) but increased the phosphorylation of AMPKα Thr^172^ and ACC Ser^79^ (Figure 3A–D and Figure 4A–E). CPT-1 expression was much lower in the OHFFD group, and 20E and PIO significantly restored the level of expression (Figure 4F–G). These findings suggest that 20E may reduce hepatic lipogenesis and increase lipolysis and/or oxidation.

### 3.4. 20-Hydroxyecdysone Reduces Fat Accumulation by Reducing Lipogenesis in White Adipose Tissue

Triglyceride synthesis is a strictly regulated process that occurs principally in adipose tissue, but also in the liver, muscle, heart, and pancreas [28]. We next investigated the mechanisms underpinning the changes in lipid metabolism in white adipose tissue related to OHFFD and the effects of 20E supplementation and PIO treatment. To this end, we measured the expression of proteins associated with lipid metabolism, including those involved in fatty acid synthesis and β-oxidation, in periovarian white adipose tissue. OHFFD rats showed significantly lower Thr^172^ phosphorylation of AMPKα and Ser^79^ phosphorylation of ACC in white adipose tissue (Figure 5A–D and Figure 6A–D), and the phosphorylation of both proteins was significantly increased by 20E supplementation, as for PIO treatment (Figure 5A–D and Figure 6A–D). There was also significantly higher expression of FAS in adipose tissue from the OHFFD group than in the Sham group, and this was reduced by 20E supplementation (Figure 6E). Moreover, the expression of the lipogenic protein sterol regulatory element-binding protein 1c (SREBP-1c) was significantly higher in the OHFFD rats, and this effect was markedly attenuated by a high dose of 20E or PIO treatment (Figure 6F–G).

## 4. Discussion

Our previous and present studies show that the long-term consumption of a high-fat, high-fructose diet causes abnormal weight gain and the development of insulin resistance, dyslipidemia, marked visceral fat deposition, and hepatic steatosis in ovariectomized rats [10,11], all of which are characteristics of obesity and MAFLD [1,2]. In the present study, we have provided new insight into the beneficial effects of 20E, a natural plant compound. We found that 20E has anti-lipogenic effects, reducing the visceral fat mass and liver fat content of a rat model of MAFLD, and we have provided evidence for the underlying mechanism whereby 20E has these effects. 

The dramatic increases in the prevalence of obesity, the metabolic syndrome, and non-alcoholic fatty liver disease have been linked to the widespread increase in the consumption of high-calorie diets, associated with greater consumption of processed foods containing saturated fat and sweetened beverages [29,30]. Rodents fed diets high in fat and fructose over a long period of time exhibit several pathological alterations in the liver that resemble those that develop in humans with metabolic dysfunction [31,32,33]. It is well documented that hepatic fat accumulation results from an imbalance between lipid accumulation and disposal [25,26,27]. Excessive lipid accumulation occurs in liver cells when the rates of delivery of free fatty acids to the liver and de novo lipogenesis (DNL) (also referred to as de novo fatty acid synthesis) exceed those of fatty acid oxidation and release [34,35]. In the liver, DNL is the key pathway involved in lipid storage, and requires the enzymes ATP-citrate lyase (ACL) and acetyl-CoA carboxylase (ACC), and the multi-enzymatic complex fatty acid synthase (FAS). ACC is the first rate-limiting enzyme in the de novo fatty acid synthesis pathway. It catalyzes the carboxylation of acetyl-CoA to produce malonyl-CoA, which is in turn used by FAS to produce long-chain saturated fatty acids [36,37]. In the present study, we found that prolonged high-fat, high-fructose diet-feeding and the loss of ovarian hormones owing to ovariectomy resulted in higher expression of both ACC and FAS, suggesting that OHFFD promotes lipogenesis in the liver. It should be noted that AMPK is an energy sensor and an important regulator of ACC1 [38]. When AMPK is activated, it phosphorylates the serine-79 residue of ACC1 [39], which inhibits the formation of the ACC1 homodimer, rendering it unable to catalyze acetyl-CoA carboxylation and reducing fatty acid synthesis. Consistent with this, AMPK was activated in OHFFD rats treated with 20E, and this was accompanied by an increase in the phosphorylation of ACC (Ser^79^), implying that 20E activates AMPK and reduces hepatic steatosis through the suppression of de novo lipogenesis. Of note, the role of fibroblast growth factor 21 (FGF21), a metabolic hormone mainly expressed in the liver, as a metabolic regulator of lipid and energy homeostasis has been reported through activation of the AMPK signaling pathway [40,41,42]. We have previously reported that the expression level of FGF21 was increased in the 20E treatment group [11]. Taken together, the underlying mechanisms of 20E to mitigate hepatic steatosis may also involve the activation of FGF21, which contributes to the suppression of DNL via AMPK and ACC pathway. In addition, the much lower expression of FAS following 20E treatment would also reduce lipogenesis, causing a reduction in the triglyceride content of the liver. Furthermore, greater β-oxidation of free fatty acids, associated with an increase in the expression of CPT-1 caused by 20E treatment, would also reduce hepatic lipid accumulation. These findings imply that 20E increases lipid β-oxidation and reduces DNL, thereby ameliorating hepatic steatosis in OHFFD rats. 

Several lines of evidence and our previous work indicate that ovariectomy or loss of ovarian hormone, especially estrogen (17β-estradiol), produced hyperphagia and increased gain in body weight and fat mass [43]. In the context of energy balance, estrogen modulates energy homeostasis by reducing food intake and increasing energy expenditure and this regulation is primarily through estrogen receptor-α (Erα)-mediated mechanisms. ER-α in hypothalamic steroidogenic factor-1 (SF1) neurons is required to regulate energy expenditure and fat distribution, and ER-α in hypothalamic pro-opiomelanocortin (POMC) neurons is required for the regulation of feeding [44,45,46]. In our previous study, we demonstrated the important role of ovarian hormones in the development of the metabolic syndrome. Ovariectomized rats develop body weight gain, central obesity, and insulin resistance [43], and the OHFFD rats used in the present study mimic the phenotype of menopausal women with MAFLD and the excessive accumulation of visceral fat. Obesity is a manifestation of the excessive expansion of white adipose tissue and fat cell hypertrophy, and the hypertrophy of adipocytes is typically associated with insulin resistance [47,48]. We previously showed that the adipocytes of ovariectomized rats are hypertrophic, especially in their visceral adipose tissue [49]. Although we did not measure the size of the adipocytes of OHFFD rats in the present study or previously, we found that OHFFD rats are characterized by substantial abdominal fat mass and insulin resistance [11]. Therefore, it is likely that OHFFD rats have hypertrophic adipocytes and that 20E would reduce this hypertrophy and ameliorate insulin resistance in these rats, as reported previously [11].

It has been reported that white adipose tissue also regulates glucose homeostasis by storing excess energy within lipid droplets of adipocytes, thus preventing insulin resistance [50,51]. PIO is a thiazolidinedione drug that is used as an insulin sensitizer in patients with type 2 diabetes mellitus. It causes the recruitment of small new adipocytes that can more effectively store excess dietary lipids and thereby ameliorate insulin resistance [52,53]. In the present study, we compared the effects of 20E administration with those of PIO. In a previous study, we showed that 20E improved the insulin sensitivity of OHFFD rats [11], suggesting that 20E may recruit small new adipocytes or prevent the hypertrophy of their adipocytes. PIO is a ligand for peroxisome proliferator-activated receptor gamma (PPAR-γ), which is a key transcription factor in the regulation of adipogenesis and stimulates adipogenesis, particularly in the subcutaneous, rather than the visceral, adipose tissue compartment. However, the underlying mechanisms of the accumulation of the visceral fat thus far have not been fully investigated. We have mentioned that the lower estrogen concentrations following menopause are associated with greater adiposity and visceral fat accumulation. This is because estrogen reduces food intake, whereas ovariectomy leads to hyperphagia. Therefore, in addition to the long-term consumption of a high-fat, high-fructose diet, the loss of ovarian hormones may also explain the abdominal fat accumulation of the OHFFD rats.

Typically, lipid accumulation in each adipose depot reflects the balance between triglyceride synthesis and lipolysis. Fatty acid synthesis in adipocytes is primarily mediated by ACC and FAS, and we have shown that the expansion of visceral fat in the OHFFD rats is accompanied by higher adipose ACC and FAS expression. Consistent with this, the protein expression and enzyme activity of ACC are upregulated in conditions of nutrient and energy abundance [54]. The activation of AMPK inhibits ACC activity, and we found that both 20E and PIO increased the phosphorylation of AMPKα (Thr^172^) and ACC (Ser^79^) in adipose tissue, implying the inhibition of lipogenesis in adipose tissue in the OHFFD rats. Also consistent with this, 20E and PIO reduced FAS expression, implying a suppression of lipogenesis in white adipose tissue, which would contribute to the lower visceral adipose mass in OHFFD rats. The lower fat mass in 20E-treated OHFFD rats could also be attributed to lower expression of a transcription factor that regulates the expression of ACC, and SREBP-1c increases the transcription of ACC and FASN, which are involved in fatty acid and phospholipid synthesis [55]. In general, SREBP-1c increases the transcription of genes that promote adipogenesis and lipid storage. Thus, low expression of SREBP-1c results in low expression of ACC, which leads to a reduction in de novo lipogenesis. We also found that 20E, in addition to PIO, reduces SREBP-1c expression in adipocytes, which may underpin the reduction in visceral fat mass. Finally, we have shown that 20E has similar effects to PIO to reduce visceral fat mass and has similar effects on the molecular mediators of fat synthesis and lipolysis. However, whether 20E may be a ligand for PPAR-γ requires further study.

One of the concerns regarding the use of natural medicines or compounds derived from medicinal plants is their toxicity. We have shown that 20E has beneficial effects in vivo, but we must also consider whether 20E or a metabolite may have toxic effects. 20E has been investigated for its toxicity in several previous studies and found to have a good safety profile. The oral LD50 of 20E was found to be >9000 mg/kg in mice, and during the long-term feeding of rats, no subacute toxicity at doses of 200–2000 mg/kg/day have been identified [56]. In addition, no detrimental effects have been reported in rat models of estrogen deprivation or ovariectomy involving feeding with 20E at doses between 18 and 121 mg/kg/day for 3 months [57,58]. Importantly, the doses of 20E (5, 10, and 20 mg/kg/day) used in the present study were lower than the reported LD50, which implies that 20E could be an effective and safe food supplement for use in patients with MAFLD.

## 5. Conclusions

We have shown that 20E, a natural compound, reduces lipid accumulation in the liver and visceral adipose tissue of OHFFD rats through the suppression of DNL and an increase in lipid β-oxidation as summarized in Figure 7. Thus, 20E ameliorates the metabolic dysfunction of an animal model of MAFLD. Given the current limited therapeutic options for patients with MAFLD/non-alcoholic fatty liver disease, these findings suggest the potential for the use of 20E to ameliorate the symptoms of MAFLD through its anti-lipogenic effects. Dietary supplementation with 20E may represent an alternative therapeutic option for the treatment and prevention of the metabolic complications associated with MAFLD, and could alleviate the related epidemics of MAFLD and its sequelae, which have become a worldwide threat. Nevertheless, the long-term effects of 20E administration require further studies to ensure the safety and effectiveness of the use of 20E for humans.

## Figures and Tables

**Figure 1 biomedicines-11-02071-f001:**
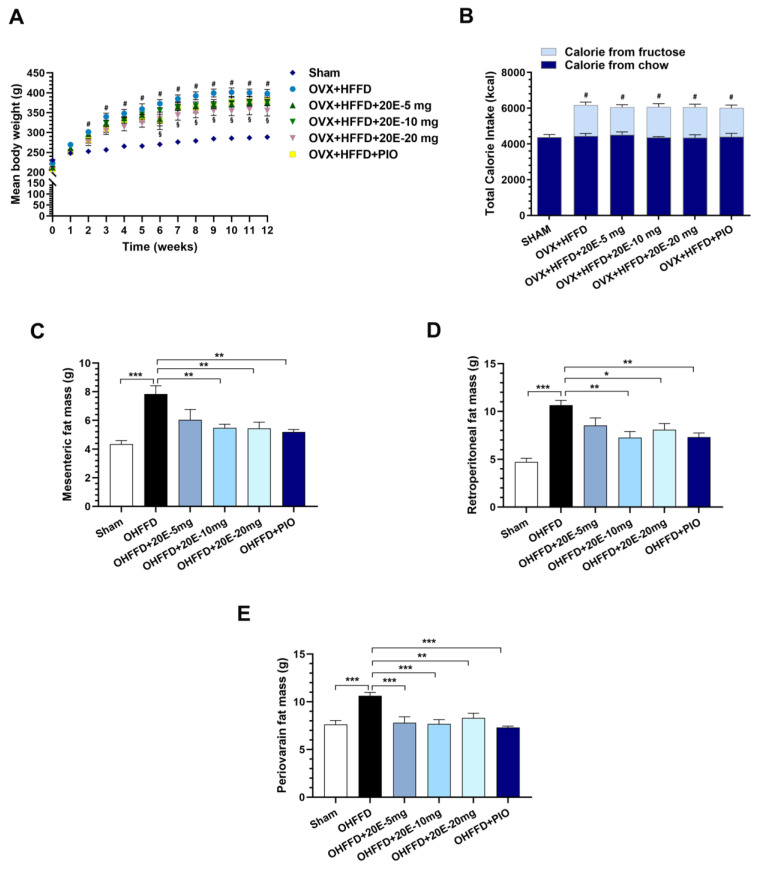
Effects of 20-hydroxyecdysone (20E) and pioglitazone (PIO) treatment on body weight, total calories intake, and visceral fat mass in ovariectomized rats fed a high-fat, high-fructose diet (HFFD). The (**A**) Mean weekly body mass, (**B**) Total caloric intake, (**C**) Mesenteric fat pad mass, (**D**) Retroperitoneal fat pad mass, and (**E**) Periovarian fat pad mass of sham-operated rats (Sham) fed a control diet, ovariectomized (OVX) rats fed a high-fat, high-fructose diet (OHFFD), OVX rats fed an HFFD and administered 20-hydroxyecdysone (OHFFD + 20E) at a dose of 5, 10, or 20 mg/kg, and OVX rats fed an HFFD and administered pioglitazone (OHFFD + PIO). Data are mean ± SEM for six animals/group. ^#^
*p* < 0.0001 (Sham vs. OHFFD); ^§^
*p* < 0.05 (OHFFD vs. OHFFD + 20E). * *p* < 0.05, ** *p* < 0.01, and *** *p* < 0.001 between the indicated groups.

**Figure 2 biomedicines-11-02071-f002:**
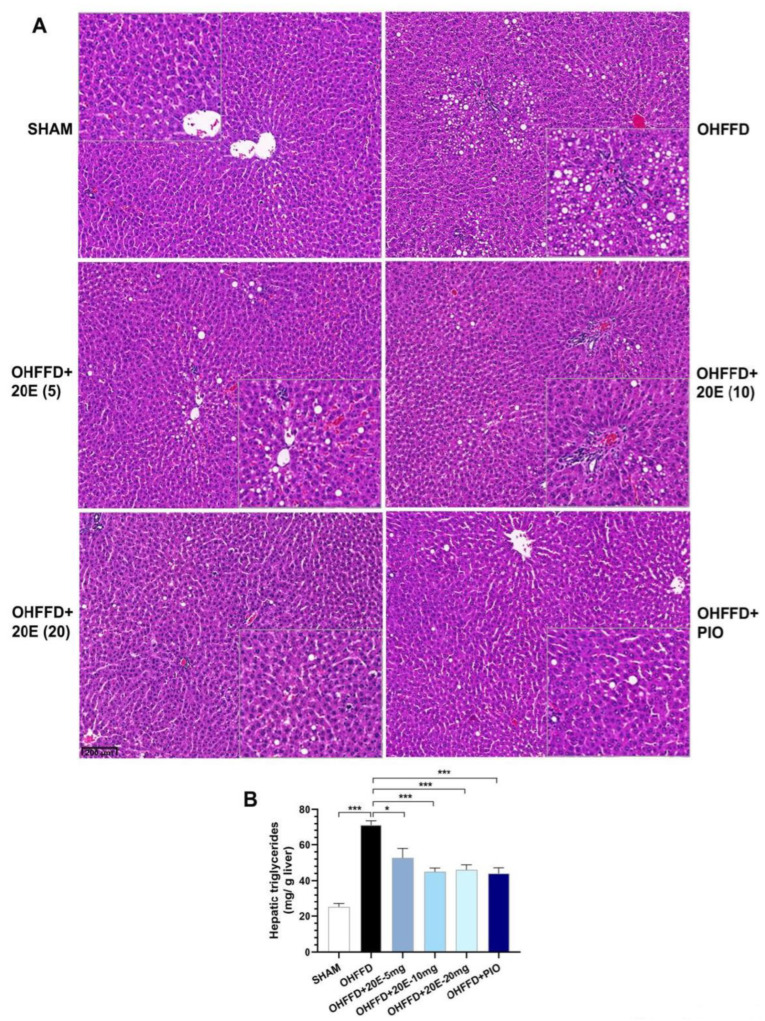
Effect of 20-hydroxyecdysone (20E) and pioglitazone (PIO) treatment on hepatic fat accumulation and hepatic triglyceride content in ovariectomized rats fed a high-fat, high-fructose diet (HFFD). (**A**) Representative photomicrographs of livers stained with hematoxylin and eosin (H&E) from sham rats fed a control diet (SHAM), ovariectomized rats fed an HFFD (OHFFD), OVX rats fed an HFFD and administered 20-hydroxyecdysone at a dose of 5, 10, or 20 mg/kg (OHFFD + 20E (5), OHFFD + 20E (10), and OHFFD + 20E (20), respectively), and OVX rats fed an HFFD and administered pioglitazone (OHFFD + PIO). Scale bar: 20 μm. (**B**) Hepatic triglyceride content. Data are mean ± SEM for six animals/group. * *p* < 0.05 and *** *p* < 0.001 between the indicated groups.

**Figure 3 biomedicines-11-02071-f003:**
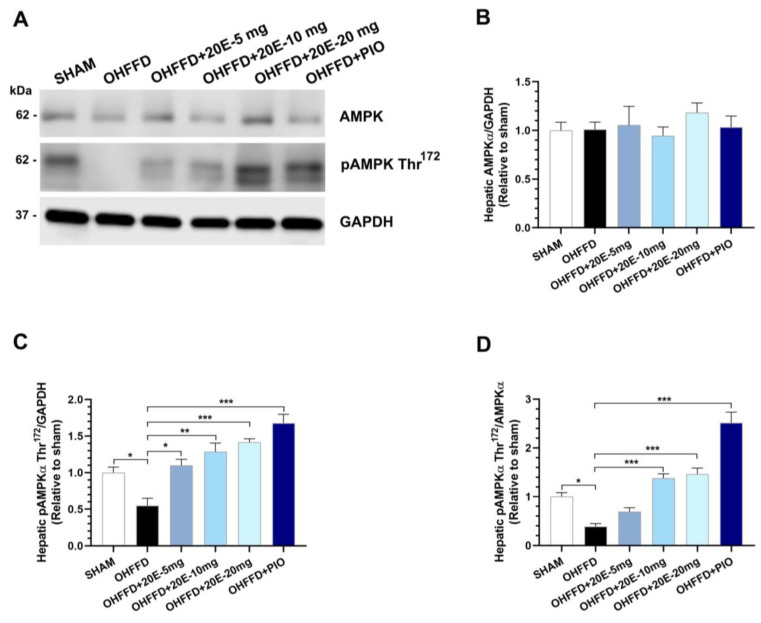
Effects of 20-hydroxyecdysone (20E) and pioglitazone (PIO) treatment on hepatic AMP-activated protein kinase (AMPK) activity. (**A**) Representative immunoblots of total AMPKα, phosphorylated AMPKα (Thr^172^) and GAPDH. The blots were cropped, and full-length blots are presented in Appendix A. (**B**) Quantitative blot analysis of total AMPKα expression, normalized to GAPDH, (**C**) Quantitative blot analysis of phosphorylated AMPKα (Thr^172^), normalized to GAPDH, and (**D**) Ratio of phosphorylated AMPKα (Thr^172^) to total AMPKα expression (pAMPKα (Thr^172^)/AMPKα), which represents AMPK activity, in sham-operated rats (Sham) fed a control diet, ovariectomized (OVX) rats fed a high-fat, high-fructose diet (OHFFD), OVX rats fed an HFFD and administered 20-hydroxyecdysone (OHFFD + 20E) at a dose of 5, 10, or 20 mg/kg, and OVX rats fed an HFFD and administered pioglitazone (OHFFD + PIO). The band intensities for pAMPKα (Thr^172^), AMPKα, and GAPDH were measured, and the data are presented as the fold differences in expression vs. the Sham group. Data are mean ± SEM for six animals/group. * *p* < 0.05, ** *p* < 0.01, and *** *p* < 0.001 between the indicated groups.

**Figure 4 biomedicines-11-02071-f004:**
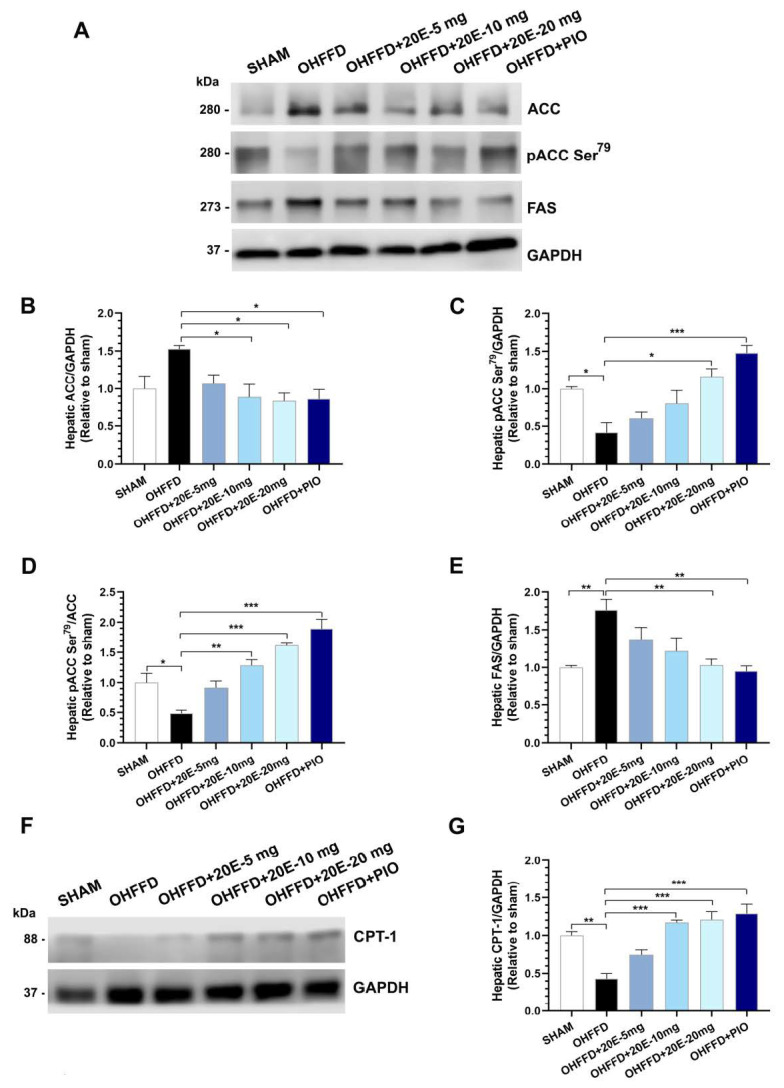
Effects of 20-hydroxyecdysone (20E) and pioglitazone (PIO) treatment on the expression of the rate-limiting enzyme of fatty acid metabolism and its regulator, and the rate-limiting enzyme of fatty acid β-oxidation in the liver. (**A**) Representative immunoblots for total ACC, phosphorylated ACC (Ser^79^), FAS, and GAPDH. The blots were cropped, and full-length blots are presented in Appendix A. (**B**) Quantitative analysis of total ACC expression, normalized to GAPDH, (**C**) Quantitative analysis of phosphorylated ACC (Ser^79^), normalized to GAPDH, (**D**) Ratio of phosphorylated ACC (Ser^79^) to total ACC (pACC (Ser^79^)/ACC), which represents ACC activity, (**E**) Quantitative analysis of FAS, normalized to GAPDH, (**F**) Representative immunoblots of CPT-1, normalized to GAPDH, the blots were cropped and full-length blots are presented in Appendix A. (**G**) Quantitative analysis of CPT-1, normalized to GAPDH for sham-operated rats (Sham) fed a control diet, ovariectomized (OVX) rats fed a high-fat, high-fructose diet (OHFFD), OVX rats fed an HFFD and administered 20-hydroxyecdysone (OHFFD + 20E) at a dose of 5, 10, or 20 mg/kg, and OVX rats fed an HFFD and administered pioglitazone (OHFFD + PIO). The band intensities for pACC (Ser^79^), ACC, and GAPDH were measured, and the data are presented as the fold differences in expression relative to the Sham group. Data are mean ± SEM for six animals/group. * *p* < 0.05, ** *p* < 0.01, and *** *p* < 0.001 between the indicated groups.

**Figure 5 biomedicines-11-02071-f005:**
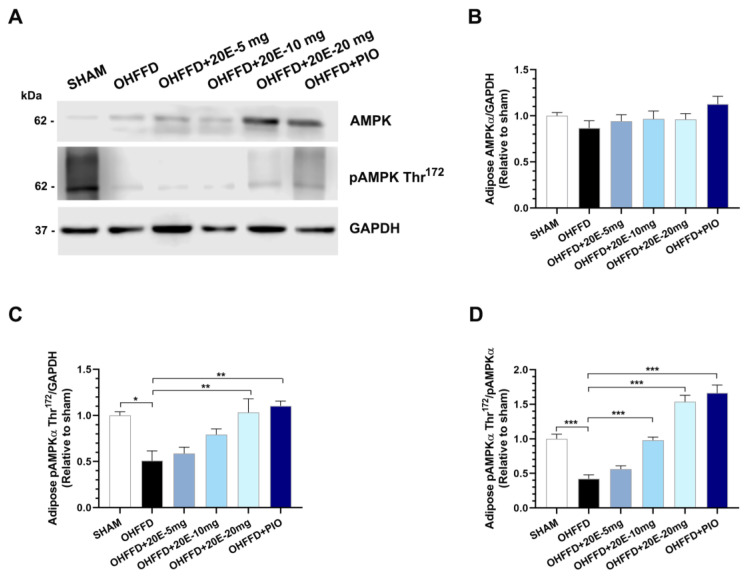
Effects of 20-hydroxyecdysone (20E) and pioglitazone (PIO) treatment on AMP-activated protein kinase (AMPK) activity in periovarian white adipose tissue. (**A**) Representative immunoblots of total AMPKα, phosphorylated AMPKα (Thr^172^), and GAPDH. The original blots are presented in Appendix A. (**B**) Quantitative analysis of total AMPKα expression, normalized to GAPDH, (**C**) Quantitative analysis of phosphorylated AMPKα (Thr^172^), normalized to GAPDH, and (**D**) Ratio of phosphorylated AMPKα (Thr^172^) to total AMPKα (pAMPKα (Thr^172^)/AMPKα), which represents AMPK activity, for sham-operated rats (Sham) fed a control diet, ovariectomized (OVX) rats fed a high-fat, high-fructose diet (OHFFD), OVX rats fed an HFFD and administered 20-hydroxyecdysone (OHFFD + 20E) at a dose of 5, 10, or 20 mg/kg, and OVX rats fed an HFFD and administered pioglitazone (OHFFD + PIO). The band intensities for pAMPKα (Thr^172^), AMPKα, and GAPDH were measured, and data are presented as the fold differences in expression relative to the Sham group. Data are mean ± SEM for six animals/group. * *p* < 0.05, ** *p* < 0.01, and *** *p* < 0.001 between the indicated groups.

**Figure 6 biomedicines-11-02071-f006:**
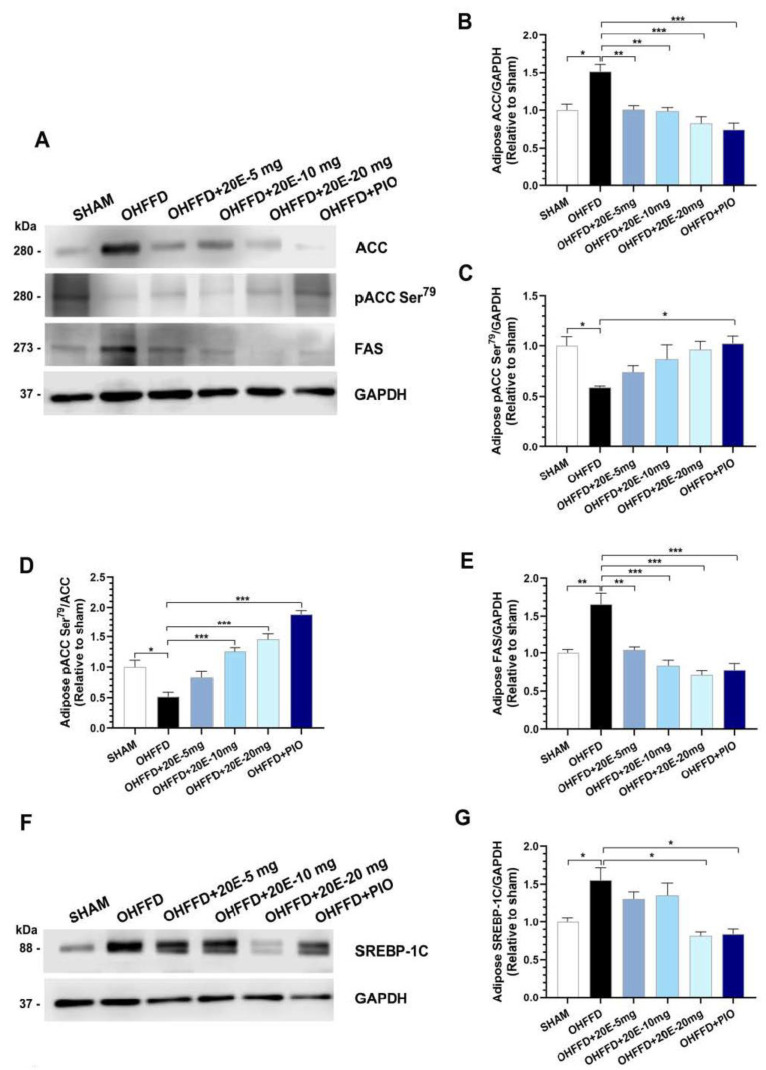
Effects of 20-hydroxyecdysone (20E) and pioglitazone (PIO) treatment on the expression of the rate-limiting enzyme in fatty acid metabolism, its regulator, and the rate-limiting enzyme of fatty acid β-oxidation in periovarian white adipose tissue. (**A**) Representative immunoblots of total ACC, phosphorylated ACC (Ser^79^), FAS, and GAPDH. The blots were cropped, and full-length blots are presented in Appendix A. (**B**) Quantitative analysis of total ACC expression, normalized to GAPDH, (**C**) Quantitative analysis of phosphorylated ACC (Ser^79^), normalized to GAPDH, (**D**) Ratio of phosphorylated ACC (Ser^79^) to total ACC (pACC (Ser^79^)/ACC), which represents ACC activity, (**E**) Quantitative analysis of FAS, normalized to GAPDH, (**F**) Representative immunoblots of SREBP-1c, normalized to GAPDH. The original blots are presented in Appendix A, and (**G**) Quantitative analysis of SREBP-1c, normalized to GAPDH for sham-operated rats (Sham) fed a control diet, ovariectomized (OVX) rats fed a high-fat, high-fructose diet (OHFFD), OVX rats fed an HFFD and administered 20-hydroxyecdysone (OHFFD + 20E) at a dose of 5, 10, or 20 mg/kg, and OVX rats fed an HFFD and administered pioglitazone (OHFFD + PIO). The band intensities for pACC (Ser^79^), ACC, and GAPDH are presented as the fold differences in expression relative to the Sham group. Data are mean ± SEM for six animals/group. * *p* < 0.05, ** *p* < 0.01, and *** *p* < 0.001 between the indicated groups.

**Figure 7 biomedicines-11-02071-f007:**
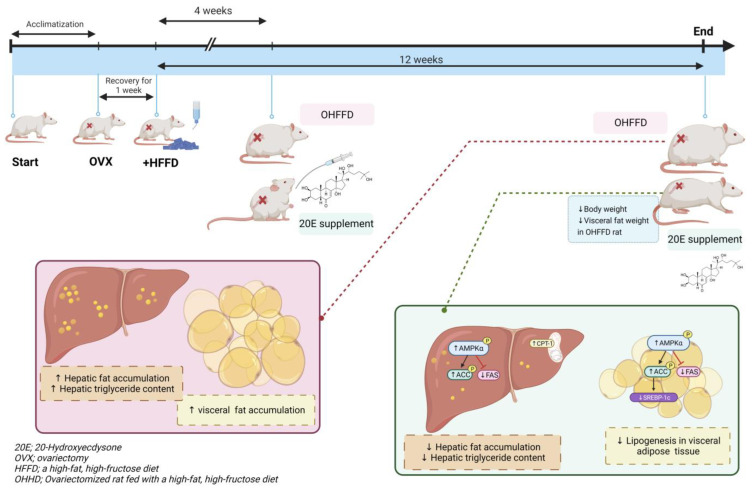
Proposed mechanism by which 20E supplementation reduces visceral fat mass and hepatic fat accumulation. Long-term high-fat-high-fructose diet and ovariectomy increase visceral and liver fat accumulation. 20E increases the phosphorylation of AMPK and ACC but decreases fatty acid synthase in the liver and adipose tissue, thereby reducing lipogenesis. 20E also increases hepatic CPT-1 and reduces the adipose SREBP-1c expression, thereby increasing fatty acid β-oxidation and reducing lipid biosynthesis and adipogenesis.

## Data Availability

The datasets used and/or analyzed during the current study are available from the corresponding author on reasonable request.

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
