# Peer review of "Dietary Supplementation with 20-Hydroxyecdysone Ameliorates Hepatic Steatosis and Reduces White Adipose Tissue Mass in Ovariectomized Rats Fed a High-Fat, High-Fructose Diet"

_biomedicines, 2023, doi:10.3390/biomedicines11072071_

Round 1

Reviewer 1 Report

Comments for biomedicines-2459612

The present study reports that 20-hydroxyecdysone (20E) improves hepatic steatosis and reduces white adipose tissue mass in a high-fat, high-fructose diet-induced ovariectomized rats model. The authors suggest that the MAFLD amelioration effects of 20E can be attributed to its influence on multiple pathways, including the AMPK signaling pathway and fatty acid synthase pathway. This study seems to specifically evaluate the effect of 20E on ameliorating metabolic disorders of menopause. However, several issues still need to be addressed, the details are as follows:

1.       Since the authors selected the determination of the AMPK pathway, why didn't they use the AMPK activator (Metformin) as a positive control instead of Pioglitazone?

2.       Could you explain why the reduction in fat mass does not show a dose-dependent manner, unlike the H&E histopathological findings?

3.       The authors should provide observations of oil red O/Nile red staining in liver tissue.

4.       All serum transaminase levels and lipid profiling, which are used to assess liver function and lipid metabolism, are missing.

5.       The previous study already mentioned that 20E activates hepatic AMPK in the same rat model. Combined the reference 11, several key proteins were investigated. So,   comparing these results, which mechanism plays the main role in the amelioration of MAFLD?

Reference 11: Buniam, J., Chukijrungroat, N., Rattanavichit, Y. et al. 20-Hydroxyecdysone ameliorates metabolic and cardiovascular dysfunction in high-fat-high-fructose-fed ovariectomized rats. BMC Complement Med Ther 20, 140 (2020). https://doi.org/10.1186/s12906-020-02936-1

Author Response

Response to Reviewer 1 Comments

The present study reports that 20-hydroxyecdysone (20E) improves hepatic steatosis and reduces white adipose tissue mass in a high-fat, high-fructose diet-induced ovariectomized rat model. The authors suggest that the MAFLD amelioration effects of 20E can be attributed to its influence on multiple pathways, including the AMPK signaling pathway and fatty acid synthase pathway. This study seems to specifically evaluate the effect of 20E on ameliorating metabolic disorders of menopause. However, several issues still need to be addressed, the details are as follows:

1. Since the authors selected the determination of the AMPK pathway, why didn't they use the AMPK activator (Metformin) as a positive control instead of Pioglitazone?

Response 1: We thank the reviewer for pointing this out. To our knowledge, lipid metabolism in the liver could be regulated by multiple pathways including the AMPK, fatty acid synthase pathway, and others. The reason that we chose Pioglitazone as a positive control is due to its primary effect on improving whole-body insulin sensitivity and its specific impact on lipid metabolism, particularly in the liver. Furthermore, evidence from meta-analysis studies [Blazina & Selph, 2019] has demonstrated that thiazolidinediones, such as Pioglitazone, exhibit more favorable effects on hepatic steatosis in patients with non-alcoholic steatohepatitis (NASH) and non-alcoholic fatty liver disease (NAFLD) compared to other insulin-sensitizing medications like metformin.

Reference:

Blazina, I.; Selph, S. Diabetes drugs for nonalcoholic fatty liver disease: a systematic review. Syst Rev 2019, 8, 295, doi:10.1186/s13643-019-1200-8.

2. Could you explain why the reduction in fat mass does not show a dose-dependent manner, unlike the H&E histopathological findings?

Response 2: We thank the reviewer for raising this point. We agree that the effect of 20E on a reduction in hepatic triglyceride content does not show a dose-dependent fashion while the representatives for liver histopathological findings tend to show a dose-dependent response. Thus, we have removed the word “a dose-dependent manner” from the sentence in the Results section to be more circumspect about what we are showing in the revised manuscript.

3. The authors should provide observations of oil red O/Nile red staining in liver tissue.

Response 3: We appreciate your constructive suggestion. We agree that the Oil red O/Nile red staining in liver tissue would provide additional evidence. However, we have quantitatively determined the level of hepatic triglyceride content to support our H&E observation. We hope that in this result the highly significant (p<0.001) difference in hepatic triglyceride content is meaningful and sufficient to convincingly make the point.

4. All serum transaminase levels and lipid profiling, which are used to assess liver function and lipid metabolism, are missing.

Response 4: We thank the reviewer for this comment. The serum transaminase levels and lipid profiling has not been incorporated in the manuscript because we have previously reported the effect of 20E on serum lipid [Buniam et al, 2020]. We, therefore, did not repeatedly measure lipid profiles in the present study. Regarding the serum transaminase levels, we have also demonstrated in a previous study [Chukijrungroat et al, 2017] that the serum ALT level, an indicator of hepatic injury, was increased only in male rats fed the HFFD, whereas no significant difference in the serum ALT level was observed in the female and OVX groups fed the HFFD. These results indicate that female rats are not vulnerable to liver damage in response to chronic HFFD feeding. Despite the above-mentioned information, the serum ALT level was determined in this study. Consistently, we found that the serum ALT level was not significantly different among any groups, so we decided not to incorporate this set of data in the manuscript.

References:

Buniam, J.; Chukijrungroat, N.; Rattanavichit, Y.; Surapongchai, J.; Weerachayaphorn, J.; Bupha-Intr, T.; Saengsirisuwan, V. 20-Hydroxyecdysone ameliorates metabolic and cardiovascular dysfunction in high-fat-high-fructose-fed ovariectomized rats. BMC Complement Med Ther 2020, 20, 140, doi:10.1186/s12906-020-02936-1.

Chukijrungroat, N.; Khamphaya, T.; Weerachayaphorn, J.; Songserm, T.; Saengsirisuwan, V. Hepatic FGF21 mediates sex differences in high-fat high-fructose diet-induced fatty liver. Am J Physiol Endocrinol Metab 2017, 313, E203-E212, doi:10.1152/ajpendo.00076.2017.

5. The previous study already mentioned that 20E activates hepatic AMPK in the same rat model. Combined the reference 11, several key proteins were investigated. So, comparing these results, which mechanism plays the main role in the amelioration of MAFLD?

Response 5: We thank the reviewer for raising this question. Referring to reference #11 [Buniam et al, 2020], Buniam et al. have examined the effect of 20E on insulin sensitivity. We found that 20E treatment effectively improves insulin sensitivity by controlling glucose homeostasis, which is associated with increasing expression of pAkt Ser473, pFOXO1 Ser256, and FGF21 in the liver tissue. Among these proteins, FGF21 is a metabolic hormone mainly expressed in the liver and functions as a metabolic regulator of glucose, lipid, and energy homeostasis [Cuevas-Ramos et al, 2012; Kralisch et al, 2011]. It has been reported that FGF21 may regulate energy homeostasis through the activation of the AMPK signaling pathway [Chau et al, 2010]. Given this information together with the results presented in this study, it is conceivable that the underlying mechanisms of the favorable effect of 20E may involve the activation of FGF21, which contribute to the suppression of de novo lipogenesis via AMPK and ACC pathway and mitigation of hepatic steatosis. In agreement with the reviewer’s suggestion, we have addressed this issue in the second paragraph of the Discussion section.

References:

Buniam, J.; Chukijrungroat, N.; Rattanavichit, Y.; Surapongchai, J.; Weerachayaphorn, J.; Bupha-Intr, T.; Saengsirisuwan, V. 20-Hydroxyecdysone ameliorates metabolic and cardiovascular dysfunction in high-fat-high-fructose-fed ovariectomized rats. BMC Complement Med Ther 2020, 20, 140, doi:10.1186/s12906-020-02936-1.

Cuevas-Ramos, D.; Aguilar-Salinas, C.A.; Gomez-Perez, F.J. Metabolic actions of fibroblast growth factor 21. Curr Opin Pediatr 2012, 24, 523-529, doi:10.1097/MOP.0b013e3283557d22.

Kralisch, S.; Fasshauer, M. Fibroblast growth factor 21: effects on carbohydrate and lipid metabolism in health and disease. Curr Opin Clin Nutr Metab Care 2011, 14, 354-359, doi:10.1097/MCO.0b013e328346a326.

Chau, M.D.; Gao, J.; Yang, Q.; Wu, Z.; Gromada, J. Fibroblast growth factor 21 regulates energy metabolism by activating the AMPK-SIRT1-PGC-1alpha pathway. Proc Natl Acad Sci U S A 2010, 107, 12553-12558, doi:10.1073/pnas.1006962107.

Reviewer 2 Report

This investigation is interesting. There are some requests to make the manuscript more better. The mechanism how the ovariectomized rats used in this study develops MAFLD-like disorder including body weight gain and fat accumulation should be added. In addtion, can upregulation or down regulation of FAS, ACC and CPT1  be recognized as main causes of fat accumulation in this model? And based on this point of view, whether 20-hydroxyecdysone can suppress fat accumulation in other MAFLD models and human MAFLD should be discuss more in detail.

It's OK with minor spell checking only.

Author Response

Response to Reviewer 2 Comments

1. This investigation is interesting. There are some requests to make the manuscript more better. The mechanism how the ovariectomized rats used in this study develops MAFLD-like disorder including body weight gain and fat accumulation should be added.

Response 1: We very much appreciate your positive review of the manuscript. To address the mechanism of how the ovariectomized rats used in our present study develop body weight gain and fat accumulation, we have included a brief discussion in the third paragraph of the Discussion.

“Several lines of evidence and our previous work indicate that ovariectomy or loss of ovarian hormone, especially estrogen (17b-estradiol) produced hyperphagia and increased gain in body weight and fat mass [7]. In the context of energy balance, estrogen modulates energy homeostasis by reducing food intake and increasing energy expenditure and this regulation is primarily through estrogen receptor-a (ERa)-mediated mechanisms. ER-a in hypothalamic steroidogenic factor-1 (SF1) neurons is required to regulate energy expenditure and fat distribution, and ER-a in hypothalamic pro-opiomelanocortin (POMC) neurons is required for the regulation of feeding [8-10].

References:

[7] Saengsirisuwan, V.; Pongseeda, S.; Prasannarong, M.; Vichaiwong, K.; Toskulkao, C. Modulation of insulin resistance in ovariectomized rats by endurance exercise training and estrogen replacement. Metabolism 2009, 58, 38-47, doi:10.1016/j.metabol.2008.08.004.

[8] Mauvais-Jarvis, F.; Clegg, D.J.; Hevener, A.L. The role of estrogens in control of energy balance and glucose homeostasis. Endocr Rev 2013, 34, 309-338, doi:10.1210/er.2012-1055.

[9] Xu, Y.; Lopez, M. Central regulation of energy metabolism by estrogens. Mol Metab 2018, 15, 104-115, doi:10.1016/j.molmet.2018.05.012.

[10] Xu, Y.; Nedungadi, T.P.; Zhu, L.; Sobhani, N.; Irani, B.G.; Davis, K.E.; Zhang, X.; Zou, F.; Gent, L.M.; Hahner, L.D.; et al. Distinct hypothalamic neurons mediate estrogenic effects on energy homeostasis and reproduction. Cell Metab 2011, 14, 453-465, doi:10.1016/j.cmet.2011.08.009.

2. In addition, can upregulation or down regulation of FAS, ACC and CPT1 be recognized as main causes of fat accumulation in this model?

Response 2: We thank the reviewer for this interesting question. It is recognized that increased fat accumulation can be caused by an imbalance between lipogenesis and fatty acid oxidation. ACC and FAS are the key lipogenic proteins while CPT-1 is a key protein in the fatty acid oxidation process. Therefore, an upregulation of ACC and FAS and a downregulation of CPT-1 can be recognized as the main causes of fat accumulation in this model. We have described the role of these proteins in the second paragraph of the Discussion section.

3. And based on this point of view, whether 20-hydroxyecdysone can suppress fat accumulation in other MAFLD models and human MAFLD should be discuss more in detail.

Response 3: We appreciate your constructive comments. Because the study of 20E in MAFLD has not been extensively investigated, it remains premature to conclude whether 20E can suppress fat accumulation in other MAFLD models and patients with MAFLD. We have addressed this point in the Conclusion section.

4. Comments on the Quality of English Language

It's OK with minor spell checking only.

Response 4: We appreciate your comments. We have made corrections accordingly.

Reviewer 3 Report

Fig.1, Fig.2 The title of the figure should be changed. The title should reflect the key information presented in the figure

Fig.2 – there is no difference in hepatic triglyceride content between groups supplemented with 10mg and 20 mg of 20E, and difference between group supplemented with 5 mg of 20E and 10 mg or 20 mg is quite small. Therefore, it is not proper to say that “hepatic triglyceride content (Figure 2B), and 20E supplementation significantly reduced this in a dose-dependent manner”.

Fig. 3 – the effect on AMPK not AMPK activation is presented, higher protein level may only suggest enhanced activity

Fig.5 – data presented on fig 5G do not correspond to data presented on fig 5G. The band intensity seems to very similar.

Line 336-337 – “AMPK was activated …” – The activity was not measured, therefore it seems that AMPK was activated as an increase in the phosphorylation of ACC was observed.

It is hard to understand the meaning of this sentence” To date, the underlying mechanisms of the accumulation of the abdominal visceral fat have not been fully characterized” ?

“This is because estrogen reduces food intake” -  The effect of estrogens and estrogen deficiency on food intake and development of obesity according to the available evidence seems to be more complex. Therefore an improvement is needed

The presented data showing the potential beneficial effect of  20E intake does not justify the claim that 20E can be used as a dietary supplement and is safe for patients. Further animal studies are needed to evaluate the potential long-term effects of 20E administration. Such data will allow to determine the possibility of using 20E in humans, and clinical trials must document the safety and effectiveness.

Author Response

Response to Reviewer 3 Comments

1. Fig.1, Fig.2 The title of the figure should be changed. The title should reflect the key information presented in the figure

Response 1: We appreciate this helpful suggestion. We have revised the title of Figures 1 and 2, as suggested.

2. Fig.2 – there is no difference in hepatic triglyceride content between groups supplemented with 10mg and 20 mg of 20E, and difference between group supplemented with 5 mg of 20E and 10 mg or 20 mg is quite small. Therefore, it is not proper to say that “hepatic triglyceride content (Figure 2B), and 20E supplementation significantly reduced this in a dose-dependent manner”.

Response 2: As also suggested by another reviewer 1, to address this, we have removed the word “a dose-dependent manner” from the revised manuscript.

3. Fig. 3 – the effect on AMPK not AMPK activation is presented, higher protein level may only suggest enhanced activity

Response 3: We thank the reviewer for this important suggestion. As the ratio of phosphorylated AMPK-a over total AMPK-a (pAMPK-a/AMPK-a) represents AMPK activity, we have reworded the sentence in the title of Figures 3 and 5 as suggested.

4. Fig.5 – data presented on fig 5G do not correspond to data presented on fig 5G. The band intensity seems to very similar.

Response 4: We thank the reviewer for pointing this out and for the opportunity to explain. Because there is no “Figure G” in Figure 5, we presume the reviewer instead refers to Figure 6F and Figure 6G which showed the representative immunoblot of SREBP-1c normalized to GAPDH and the quantitative analysis of protein expression of SREBP-1c in periovarian white adipose tissue. The explanation for the quantitative analysis (band intensity) of protein SREBP-1c in Figure 6G that seem not to correspond to the representative blot presented in Figure 6F is due to the variation of quantitative analysis of protein derived from multiple immunoblots (6 independent experiments). Due to their high lipid and low protein contents, Western blot analysis of white adipose tissue is commonly susceptible to variation. We meticulously selected the representative immunoblot of SREBP-1c to closely correspond to the quantitative analysis of SREBP-1c. Although the quantitative analysis of protein SREBP-1c is not exactly perfect for the representative blot, the key finding and interpretation remain the same conclusion.

5. Line 336-337 – “AMPK was activated …” – The activity was not measured, therefore it seems that AMPK was activated as an increase in the phosphorylation of ACC was observed.

Response 5: Thanks for the comment. The reviewer asks about AMPK and ACC. Concerning the AMP-activated protein kinase (AMPK), AMPK is activated by phosphorylation of Thr172 within the activation loop of the a-subunit, and many studies have shown that Thr172 phosphorylation increases AMPK activity [Steinberg et al, 2019]. Referring to reference 39 [Steinberg et al, 2009], “When AMPK is activated, it phosphorylates the serine-79 residue of ACC, which inhibits the formation of the ACC1 homodimer, rendering it unable to catalyze acetyl-CoA carboxylation and reducing fatty acid synthesis.” This indicates that AMPK is an upstream regulator of ACC.

References:

Steinberg, G.R.; Carling, D. AMP-activated protein kinase: the current landscape for drug development. Nat Rev Drug Discov 2019, 18, 527-551, doi:10.1038/s41573-019-0019-2.

Steinberg, G.R.; Kemp, B.E. AMPK in Health and Disease. Physiol Rev 2009, 89, 1025-1078, doi:10.1152/physrev.00011.2008.

6. It is hard to understand the meaning of this sentence” To date, the underlying mechanisms of the accumulation of the abdominal visceral fat have not been fully characterized”?

Response 6: To address this concern, we have reworded the sentence to make it more comprehensible.

7. “This is because estrogen reduces food intake” -The effect of estrogen and estrogen deficiency on food intake and development of obesity according to the available evidence seems to be more complex. Therefore, an improvement is needed

Response 7: As also suggested by another reviewer, to address this, we have clarified this point in the third paragraph of the Discussion section.

8. The presented data showing the potential beneficial effect of 20E intake does not justify the claim that 20E can be used as a dietary supplement and is safe for patients. Further animal studies are needed to evaluate the potential long-term effects of 20E administration. Such data will allow to determine the possibility of using 20E in humans, and clinical trials must document the safety and effectiveness.

Response 8: We appreciate this constructive suggestion. We agree that our findings remain premature to claim that 20E is safe for patients with MAFLD. We now have included this concern in the Conclusion section of the manuscript.

Round 2

Reviewer 1 Report

According to the reviewer's suggestion, the authors give an appropriate explanation and revision for the manuscripts.